# Agro-Physiological Traits of Kaffir Lime in Response to Pruning and Nitrogen Fertilizer under Mild Shading

**DOI:** 10.3390/plants12051155

**Published:** 2023-03-03

**Authors:** Rahmat Budiarto, Roedhy Poerwanto, Edi Santosa, Darda Efendi, Andria Agusta

**Affiliations:** 1Department of Agronomy, Faculty of Agriculture, Universitas Padjadjaran, Jatinangor, Sumedang 45363, Indonesia; 2Department of Agronomy and Horticulture, Faculty of Agriculture, IPB University, Dramaga, Bogor 16680, Indonesia; 3Research Center for Pharmaceutical Ingredient and Traditional Medicine, National Research and Innovation Agency, Cibinong 16911, Indonesia

**Keywords:** *Citrus hystrix*, chlorosis, leaf production, photosynthesis, shading

## Abstract

Mild shading has been reported to increase leaf production in kaffir lime (*Citrus hystrix*) through the improvement of agro-physiological variables, such as growth, photosynthesis, and water-use efficiency; however, there is still a knowledge gap concerning its growth and yield after experiencing severe pruning in harvest season. Additionally, a specific nitrogen (N) recommendation for leaf-oriented kaffir lime is still unavailable due to its lesser popularity compared to fruit-oriented citrus. The present study determined the best pruning level and N dose based on agronomy and the physiology of kaffir lime under mild shading. Nine-month-old kaffir lime seedlings grafted to rangpur lime (*C. limonia*) were arranged in a split-plot design, i.e., N dose as a main plot and pruning as a subplot. Comparative analysis resulted in 20% higher growth and a 22% higher yield in the high-pruned plants by leaving 30 cm of main stem above the ground rather than short ones with a 10 cm main stem. Both correlation and regression analysis strongly highlighted the importance of N for leaf numbers. Plants treated with 0 and 10 g N plant^−1^ experienced severe leaf chlorosis due to N deficiency, while those treated with 20 and 40 g N plant^−1^ showed N sufficiency; thus, the efficient recommendation for kaffir lime leaf production is 20 g N plant^−1^.

## 1. Introduction

Citrus is one of the leading, popular horticultural fruit commodities commonly used for fresh food and beverages [1,2,3]. Published biogeographic, genomic, and phylogenetic analyses determine the Southeast Himalayan foothills as the center of origin for most citrus species [4]. As one of the interesting *Citrus* taxa, lime is reported to have highly polymorphic characteristics, derived from four major *Citrus*, namely *C. medica*, *C. maxima*, *C. micrantha* and *C. reticulata*) [5]. In contrast, the kaffir lime is classified as a relatively minor citrus and apparently wild, native to central Malesia or the Southeast Asian region [6]. This lime is well-designed as a leaf-oriented target due to its aromatic leaves being used as spices in numerous Asian dishes [7,8,9]. Aside from its fragrance, the bifoliate characteristics of kaffir lime leaves can be used to differentiate this species from others [10,11]. The leaf of the kaffir lime is also famous for producing essential oils [12] that possess several bioactivities, such as antifeedant [13], antibacterial [14], and larvicide [15]. Due to its importance as a spice and an essential oil raw material, the demand for kaffir lime leaves is potentially increased. Thus, some effort is required to meet those needs. General strategies to boost plant production could employ both input intensification and land expansion [16].

Kaffir lime generally grows naturally or is planted in polyculture in the yard by residents with a non-intensive cultivation system [7,9]. In nature, kaffir lime typically grows from seed and shows a high tree appearance, long and large thorns, dense canopy and branches, so leaf harvest is less effective and efficient. On the other hand, several local communities in Tulungagung, Indonesia have implemented monoculture and semi-intensive kaffir lime cultivation [9]. Previous research has also proven that modification of cultivation under low-level shading (light reduction of about 23%) can produce beneficial stress, increasing the growth rate and yield of kaffir lime leaves by 84 and 63%, respectively [17]. Although that increasing response has been reported in the first harvest period, further research is urgently needed to confirm yields in the following period.

Kaffir lime experiences heavy pruning during its harvesting season. Heavy pruning is carried out by cutting most of the canopy and leaving a portion of the main stem for successive growth in the next period [9]. The remaining part of the main stem should be studied further since no standard has been set. Based on the results of interviews with several farmers, the previous studies reported a variation in the height of the postharvest main stem of between 10 and 30 cm above ground level [9]. The difference in that height is associated with the severity level of pruning. The height of the remaining main stem should be studied further as it relates to food reserves for growth in further seasons. Pruned plants may likely experience a different source and sink balance condition than unpruned ones. Too severe pruning may result in improper vegetative growth. Earlier studies reported a vegetative growth reduction of mandarin citrus due to heavy pruning [18]. Concerning leaf-oriented citrus such as kaffir lime, vegetative growth inhibition can directly threaten yields and profits. Therefore, there is an urgency to obtain the best pruning level for kaffir lime.

In addition to the importance of determining the pruning severity, optimizing leaf production in kaffir lime agribusiness needs to be supported by the best nitrogen (N) fertilizer doses. N is the imperative macronutrient for normal plant growth and development [19]. More specifically, this nutrient is required for producing chlorophyll, proteins, nucleic acids [20], amino acids, and sugar [21]. Thus, plant productivity highly depends upon N fertilization [22,23]. Previous studies have proven the role of N in increasing the vegetative growth of Eureka lemons and Maltese sweet citrus [24]. Concerning the kaffir lime, previous studies reported a strong and positive correlation between leaf nitrogen status and leaf oil production [25], strengthening the argument that N is urgently needed to produce good quality kaffir lime leaves. Unfortunately, there is no specific N-fertilizer dosage recommendation for leaf-oriented citrus such as kaffir lime. In Indonesia, the citrus research center has issued a recommendation only for fruit-oriented citrus, with a range of 10–20 g N per plant, for 1-year-old plants [26]. This dosage can be used as a reference for compiling specific recommendations for leaf-oriented kaffir lime cases.

Interestingly, there are some knowledge gaps pertaining to kaffir lime leaf production, i.e., the confusion resulting from a variation in the height of the postharvest main stem, the lack of information on the growth and yield in the following post-pruning period under similar mild-shading conditions, and the missing specific N recommendations for leaf-oriented production. Therefore, the present study aimed to determine the best pruning level and N dose based on the agro-physiological characteristics of kaffir lime under mild shading.

## 2. Results

### 2.1. Nitrogen Status in Soil, Leaves, and Canopy N Uptake

The present experiment measured both N status in leaves and soil at 90 DAT, and the results showed no significant differences in total soil N levels under different doses of N fertilizer applied (Figure 1A). The total N content of the soil in all the treatments ranged from 0.19 to 0.20%. In contrast, N-fertilizer doses had a noticeable effect on the total N content of the leaf tissue, with the highest N–leaf tissue in the highest N-fertilizer dose (40 g plant^−1^), while the lowest N–leaf tissue in control/no N fertilizer applied. Compared to the control, the increase in N–leaf tissue was varied, by about 13% on 10 g N plant^−1^, 17% on 20 g N plant^−1^, and 34% on 40 g N plant^−1^. Similarly, the increase in N-fertilizer doses was surely followed by a significant improvement in canopy N uptake (Figure 1B).

The results of the regression analysis showed a high coefficient of determination (R^2^) of 0.99. It was likely that the N-fertilizer dose variable could be used to estimate kaffir lime canopy N uptake by employing the provided mathematical equation. Additionally, correlation analysis also confirmed (i) the insignificant and weak correlation between N–soil and N–leaf tissue; and (ii) the significant, positive, and strong correlation between N–leaf tissue and canopy N uptake (r 0.97) (Table 1). The importance of N, as represented by canopy N uptake, for plant growth and physiological processes was also proved by correlation analysis, since canopy N uptake showed significant, positive, and strong correlation to certain variables of plant growth and physiology, namely relative growth rate, shoot number, leaf numbers, plant fresh weight, and leaf chlorophyll content.

### 2.2. Growth Performance under Different Pruning Levels and N Dosages

Growth performance seemed to be significantly affected by the single factor effect, rather than a combination of both N dosage and pruning factor (Table 2). The results of the analysis of correlation highlighted the positive, strong and significant correlation between relative growth rate (RGR) to plant height (r 0.97), number of shoots (r 0.96) and number of leaves (r 0.99) (Table 1). Another statistical analysis, namely DMRT, also reported that the RGR, plant height, and number of leaves were increased significantly, along with the increase in the dose of N fertilizer given. Plants treated with 40 g N per plant showed the best growth performance, especially compared to the control, by displaying significant improvements of about 99% on RGR, 48% on plant height, and 146% on leaf numbers. However, the best growth performance on the 40 g N plant^−1^ treatment was not significantly different to the 20 g N plant^−1^.

Harvesting activity employing a high pruning type (leaving the main stem at 30 cm above the ground) stimulated greater support to subsequent kaffir lime growth, rather than the short pruning type (leaving the main stem at 10 cm above the ground). The high pruning type had 20% greater growth rate than the short ones. However, the short pruning type successfully induced more shoots than the high ones. In terms of leaf numbers, the result was not significantly different between the treatments.

### 2.3. Plant Production under Different Pruning Levels and N Dosages

The statistical analysis depicted an insignificant interaction effect of N dose and pruning on all the variables observed related to plant production. Kaffir lime production was solely influenced either by N dose or pruning. Concerning N-fertilizer dosage, the best results of plant fresh weight were observed in plants fertilized with 40 g N that experienced an increase of 55.13 g compared to control. However, it was not markedly different from those fertilized with 20 g N (Table 3). The fresh weight of plants fertilized with 10 g N was 30% higher than controls. However, it was still 25% lower than those fertilized with 40 g N. Concerning the pruning levels, the short-pruned type is thought to have a lower assimilated reserve than the high-pruned ones, leading to a lower production response. Pruning by leaving 30 cm of the main stem above the ground resulted in the improvement of plant and leaf production by about 28 and 22%, respectively, compared to the short pruning type.

The present experiment also revealed the partition of biomass in the entire plant body in response to N dose and pruning factor. In the absence of N fertilizer, the stem became the most dominant part, representing more than 50%. In the presence of N fertilizer, the more N applied, the more dominant the leaf biomass, exceeding the stem portion, in contrast to previous case (Figure 2A). Unlike the root and stem parts, the leaves are the most commercially valuable part of the kaffir plant. Therefore, the context of yield in kaffir lime is associated with the number and weight of leaves harvested in a certain unit of growing area. The estimation of leaf production extrapolated from the present findings, showed that plants fertilized with 40 g N and 20 g N received the best treatment, with an increase in yield of more than 2× compared to control (Figure 2B). In addition, regression analysis also re-confirmed and identified a strong association between (i) leaf fresh weight and N dose (Figure 3A) and (ii) leaf numbers and N dose (Figure 3B). The application of N fertilizers at various doses showed a quadratic pattern on both the fresh weight of leaves (R^2^ = 0.9864) and leaf numbers (R^2^ = 0.9998). The fresh weight of leaves in plants fertilized with 10 g N, 20 g N and 40 g N increased over controls, by 82, 174 and 198% respectively. The number of leaves in plants fertilized with 10 g N, 20 g N and 40 g N increased over controls, by 72, 123 and 146% respectively. However, there was no significant difference in fresh weight of leaves and number of leaves between 40 g N plant^−1^ and 20 g N plant^−1^.

### 2.4. Physiological Response of Kaffir Lime under Different Pruning Levels and N Dosages

The alteration of growth and production of kaffir lime in response to different N-fertilizer dosages and pruning levels was followed by the variable of plant physiology. Higher leaf production in N-fertilized plants was associated with an increase in the rate of plant photosynthesis. The rate of photosynthesis between plants fertilized with 20 g N and 40 g of N was insignificant; however, there was a noticeable increase of about 15% compared to both control (0 g N) and 10 g N (Table 4). In contrast, there was no significant difference in N-fertilizer dosage on stomatal conductance, transpiration rate, stomatal limitation to CO_2_ uptake, and water-use efficiency (WUE) in the kaffir lime plants. Stomatal conductance, transpiration rates, intrinsic WUE and instantaneous LUE in the present experiment varied in the range of 0.37–0.38 mol H_2_O m^−2^ s^−1^, 6.25–6.59 mmol H_2_O m^−2^ s^−1^, 2.68–3.23 μmol CO_2_ mmol H_2_O^−1^, and 0.58–0.98 μg lux^−1^, respectively.

As the only variable that was significantly affected by N-fertilizer dose, the rate of photosynthesis in kaffir lime was strong and positively correlated to the content of chlorophyll, i.e., chlorophyll a (r 0.86) and chlorophyll b (r 0.85) (Table 1). The chlorophyll content increased along with the increase in N fertilizer applied to kaffir lime plants and was actually supported by the morphological fact that could be seen directly in the field. Based on field observations, the yellowish color on kaffir lime leaves fertilized with 0 g N and 10 g plant^−1^ was an early symptom of N-deficiency stress. Meanwhile, plants with 20 g N and 40 g N plant^−1^ experienced normal green leaves, presumably not experiencing a N-deficiency condition (Figure 4). Such external leaf color variations between the N-fertilizer dosages used could be reconfirmed by the results of pigment analysis.

The statistical analysis of the pigment content of kaffir lime revealed the significant effect of N dosage on plant chlorophyll content, both chlorophyll α and chlorophyll β. The absence of N-fertilizer application on the control treatment showed the lowest pigment chlorophyll content of all (Table 5). The higher the N-fertilizer dose applied, the higher the content of chlorophyll, chlorophyll β and chlorophyll total in kaffir lime leaves. In the 40 g N treatment, the content of chlorophyll α and chlorophyll β and its total chlorophyll increased significantly, by about 1.37 mg g^−1^, 0.48 mg g^−1^, respectively and 1.85 mg g^−1^ compared to the lime without fertilizer.

## 3. Discussion

Pruning is the agricultural technique used to regulate plant growth, reduce pest and disease incidence, and increase horticultural management effectiveness [27,28,29,30,31]. Canopy rejuvenation is the foremost important benefit obtained from such a technique. In a rejuvenated canopy, new leaves grew to immediately restore the lost foliage [32] and these leaves possess much more productivity [33] due to the higher potential of carbon assimilation [32]. A previous study reported the use of light pruning in the form of pinching to induce robust canopy growth in early seedlings of kaffir lime [17]. In the postpruning period, the massive growth of lateral shoots is caused by lowering apical dominance [34,35,36]. In the fruit-oriented major citrus of mandarin, heavy pruning was applied to rejuvenate the canopy, but as a result, there was a decline of vegetative growth due to a severe decline in plant resource capacity [18]. Thus, mild to moderate pruning was highly recommended for those kinds of citrus [37,38]. However, to harvest the leaf yield of kaffir lime, growers have to apply heavy pruning. Due to the nature of citrus plants which are sensitive to leaf disturbance [17,18], the pruning level should be adjusted to be lighter. Comparative analysis on the pruning factor resulted in a significant increase in growth rate, plant and leaf production by about 20, 28 and 22%, respectively, in the high pruning compared to the short pruning that left only 10 cm of the main stem. The higher assimilate reserve in the existing main stem could likely support a higher growth rate, and final yield, regardless of nitrogen doses. Therefore, the recommendation for pruning in kaffir lime is the high pruning type by leaving the main stem 30 cm above the grafted join spot.

Aside from pruning, nitrogen management was also evaluated to provide the first N-fertilizer recommendations for kaffir lime grown under artificial mild shading. Most citrus growers use leaf tissue rather than soil as the basis for fertilizer application. Leaf tissue is a more representative proxy for estimating a tree’s nutritional status for mobile nutrients, such as nitrogen [39,40,41]. The result of the leaf-based nutritional test should be compared with the optimal range of that nutrient [42,43]; thus, there is an urgency to estimate the optimal range of nutrients for achieving a profitable citrus yield [44]. A previous study [45] found the variation in optimal leaf nutrient contents in four fruit-oriented popular citrus, namely oranges, mandarin, grapefruit and pomelo. The existence of variations in the optimal leaf nutrient content is thought to be related to differences in cultural practices, edaphic, climatic, and genetic factors [44]. In fact, leaf nutrient-based citrus fertilizer recommendation guidelines have been intensively studied for seventy years in the USA [46,47,48], and have been updated several times by numerous researchers [49,50,51].

Concerning leaf nitrogen, published studies have produced recommendations based on total leaf-N concentration for popular fruit-oriented citrus species [45,50,52]. Interestingly, the present study proposed the total leaf N range (2.06–2.36% N total) for the leaf-oriented minor citrus, kaffir lime, grown under mild shading. The correlation and regression analysis were adopted in the present experiment since earlier studies frequently used it to estimate the relationship between citrus yield and leaf nutrient concentrations [45,50,53]. Our findings highlighted the positive and strong correlation between leaf-N status and relative growth rate (r 0.93) and final leaf numbers harvested (r 0.93). Regression analysis also found positive quadratic patterns between (i) N-fertilizer dose and fresh weight of leaves (R^2^ 0.9864) and (ii) N-fertilizer dose and leaf numbers (R^2^ 0.9998).

Due to its quadratic pattern, a dose of 40 g N plant^−1^ does not automatically become the best fertilizer recommendation. It has already been reported that plants fertilized with both 40 g N and 20 g N doubled the yield of the control. However, 20 g N plant^−1^ is more efficient with a relatively similar effect to 40 g N plant^−1^ for increasing growth rate, photosynthetic rate, fresh weight of leaves and leaf numbers of kaffir lime under mild-shading conditions. Mild shading was previously reported to produce a beneficial stress instead of a harmful one [17]. Best practice of N-fertilizer application under lightly shaded conditions may become a combo booster for kaffir lime leaf production. Similarly, the success of N fertilizer and beneficial shading for boosting plant growth performance was also reported by previous researchers, as indicated by larger and broader leaves [54], more dominant vegetative growth [55], and higher yield [56]. In contrast, slower plant growth leading to lower production performance was observed both in the control and the 10 g N plant^−1^ treatment. Vegetative growth inhibition is a common plant response under N-deficient conditions [57].

Aside from vegetative improvement, another advantage of best N-fertilizing practice is the regulation of assimilate translocation priority. N adequacy seemed to alter the assimilate translocation priority in kaffir lime plants. A N-sufficient plant, treated with both 20 g N plant^−1^ and 40 g N plant^−1^, showed a dominant portion of leaves, while a deficient plant likely had a large portion of stem. Concerning agribusiness profit, the leaf part of the kaffir lime is more valuable and profitable than the stem or even the root [7,9,10], due to the content of various beneficial phytochemicals such as citronellal, citronellol, citronellyl acetate, linalool and caryophyllene that contribute to the strong aromatic formed [12,58,59].

The N-sufficient condition in the 20 g N plant^−1^ and 40 g N plant^−1^ displayed a good leaf–N total (>2%) associated with proper growth performance. That growth improvement is mainly caused by a higher photosynthetic rate and chlorophyll content. Similarly, the relationship between RGR to photosynthetic and chlorophyll content was confirmed by the correlation results, with coefficients of correlation of about 0.94 and 0.91, respectively. A N-sufficient status is vital for constructing optimal leaf photosynthesis. The leaf-N status represents the protein content for the Calvin cycle and thylakoids that are subsequently associated with leaf photosynthetic capacity [60]. Concerning the normal leaf cells of C_3_ plants, including kaffir lime, N is allocated mostly in the chloroplast, at about 75%, with 10% in the cell wall, and 5% in mitochondria [61]. The variation of those partitions may occur in N-deficient conditions.

The N-deficient plant, as observed in the control and 10 g N plant^−1^, displayed a low leaf–N total (<2%) and exhibited a yellow leaf appearance, implying a chlorosis phenomenon. The data of the chlorophyll test also displayed a significant reduction compared to the N-sufficient plant. Leaf chlorophyll content was previously reported to be crucial for assimilation rate since it positively and strong correlates to leaf photosynthetic rate [62]. The degradation of chlorophyll, called chlorosis, begins to appear on the lower leaves prior to spreading over the entire canopy, and even in severe cases, it can cause necrosis on old leaves [57]. Chlorosis, as a popular N-deficient symptom, is caused by a failure to form chlorophyll pigments [63]. A published study in oranges and pomelo described the main reason behind lowering CO_2_ assimilation, i.e., impairment of the thylakoid structure and photosynthetic electron transport chain (PETC) in the leaves and declining leaf photosynthetic pigment levels [64]. Moreover, N-deficient leaves are proven to have smaller chloroplasts and no starch granules. In contrast, N-sufficient leaves have large chloroplasts, with fully formed grana components and larger starch granules for performing greater assimilation [65]. In contrast, mild-shaded and N-fertilized plants may have a more robust photosynthetic apparatus, as evidenced by a large number of thylakoids per granum and an abundance of grana per chloroplast [54].

## 4. Materials and Methods

### 4.1. Study Site

The experiment was carried out at Pasir Kuda experimental field of IPB University, Bogor, Indonesia (6°36′36″ S, 106°46′47″ E, 263 m above sea level) from November 2018 to March 2019. The soil description of the Pasir Kuda experimental field was a sandy clay latosol soil with an actual pH, C-organic, N total, P total, and K total of about 6.7, 2.37%, 0.19%, 240 mg P_2_O_5_ 100 g^−1^, and 160 mg K_2_O 100 g^−1^. During the study period, the experimental field was exposed to the rainy season, with monthly rainfall intensity ranging from 230 mm to 318 mm (x-bar 289 mm).

### 4.2. Planting Materials

Plant materials were nine-month-old seedlings obtained by the grafting technique that combined kaffir lime (*Citrus hystrix* DC) scions onto rangpur lime (*C. limonia* Osbeck) rootstock. Before field transplanting, seedlings underwent initial selection to confirm only selected plants were involved in the present experiment, with certain requirements, such as bifoliate leaves, pest and disease-free, normal growth, dormant apical bud, uniform in leaf numbers (30 ± 2 leaves) and plant height (60 ± 4 cm).

### 4.3. Research Procedure

The present study employed a split-plot experimental design, with N dosage as the main plot and pruning level as the subplot. Four levels of N dosage were tested, viz., 0 g N plant^−1^, 10 g N plant^−1^, 20 g N plant^−1^, and 40 g N plant^−1^. Two levels of pruning were also evaluated, namely short and high pruning. Short pruning was technically defined as leaving only 10 cm of the main stem above the ground, whereas high pruning gave the cutting point at 30 cm from above the ground. Six replications were provided for each combination treatment; thus, 46 experimental units in the form of kaffir lime seedlings were counted in total.

Kaffir lime seedlings were raised in a monoculture cropping system under mild-shading conditions that were artificially formed by installing a black shading net 2 m above the soil surface. In a previous, similar study, this treatment resulted in (1) a reduction of sunlight, ambient temperature, and soil temperature of about 23, 6.3, and 6.5%, respectively; and (2) the improvement of ambient relative humidity by about 2%, compared to open field monoculture system [17]. Kaffir lime transplanting to the field was conducted in November 2018, with a 50 cm × 50 cm planting distance.

Pruning was applied, according to the treatment, in December 2018, or 30 days after planting (DAP). The cutting point for pruning was actually determined based on the grafted join spot. That spot was normally found 15 cm above the stem base of the rootstock variety. However, the joining spot seemed to equal the soil surface due to the soil banking technique for suppressing undesired shoot growth from the rootstock variety.

Inorganic fertilizers, apart from N, such as phosphorus (P) and potassium (K) were applied uniformly at 33 DAP, in the form of 15 g P_2_O_5_ and 10 g K_2_O, following the national citrus agency recommendations [26]. N fertilization was carried out simultaneously with P and K, with a dose adjusted to the treatment. All mentioned fertilizer was delivered in the morning (7.00 am) of a sunny day through a soil drench surrounding the seedling (10 cm away from the main stem). Hand-weeding was routinely applied every month. Pest and disease inspection was conducted weekly, and the damage was chemically managed. Harvesting was scheduled at 120 DAP in March 2019.

### 4.4. Measured Variables

Measured variables were N status in soil and leaves, canopy N uptake, growth performance, plant production (fresh weight), and physiological responses. The N content in the soil and leaf samples was analyzed by using Kjeldahl method at 115 DAP. Plant production (fresh weight) was measured on harvesting day (120 DAP) by weighing either the whole plant or individual parts in the analytical balance (Hwh, China). Plant samples were then dried using an oven at 80 °C for 3 days to obtain plant dry weight by using a similar analytical balance. Canopy N uptake was obtained from the multiplication of the canopy dry weight and the N content of leaves. The relative growth rate (RGR) was calculated based on the ratio of the increase in plant dry weight to the number of weeks from the 1st pruning (at 30 DAP) to the 2nd pruning (120 DAP), i.e., 13 weeks. Plant height, shoot number, and leaf numbers were also observed on harvest day using a roll meter (Kenmaster, North Jakarta, Indonesia), and hand counter (Kenko, North Jakarta, Indonesia), respectively.

Plant physiological variables such as photosynthetic rate (μmol CO_2_ m^−2^ s^−1^) and transpiration rate (mmol H_2_O m^−2^ s^−1^) were measured at 94 HSP at 10.00 am (sunny day) using the Li-6400XT portable photosynthesis system (Licor Inc, Lincoln, NE, USA). In addition, the present experiment also measured (i) intrinsic water-use efficiency (μmol CO_2_ mmol H_2_O^−1^) and (ii) instantaneous LUE (μg lux^−1^) by (i) dividing the rate of photosynthesis by the rate of transpiration rate and (ii) dividing the fresh weight of leaves by the perceived sunlight amount, respectively. The amount of perceived sunlight was measured by Lux-28 portable digital lux meter (Danoplus, Hong Kong, China). Leaf pigment was measured at 86 DAT in the present experiment by following the Sims and Gamon method [66].

### 4.5. Data Analysis

Quantitative data obtained in the present experiment was subjected to the analysis of variance (ANOVA) and any significant difference found was further analyzed by the Duncan’s multiple range test (DMRT) at α 0.05. Pearson correlation analysis was carried out to find the closeness of the relationship between the observed variables, such as N status, plant growth, production, and physiological response. In addition, regression analysis was also performed to elucidate the association between N dose and certain important yield-related variables, such as canopy N uptake, fresh weight of leaves and leaf numbers. All statistical analysis was performed using the Statistical Tool for Agricultural Research (STAR) version 2.0.1.

## 5. Conclusions

The present experiment succeeded in determining the best practice for pruning and N fertilizer to boost leaf production under mild-shading conditions. Pruning by leaving 30 cm of the main stem above the ground resulted in a significant improvement of plant and leaf production by about 28 and 22%, respectively, compared to the short pruning that left only 10 cm of the main stem. The higher assimilate reserve in the existing main stem likely produced higher support for recovering the lost foliage. Concerning N management, N-fertilizer dosage had a noticeable effect on the total N content of the leaf tissue, with the highest N–leaf tissue content achieved from the highest N dose. Both correlation and regression analysis confirmed that N is crucial for plant growth and plant yield due to the role of this nutrient in chlorophyll content and photosynthetic rate. Kaffir lime treated with 0 and 10 g N plant^−1^ experienced N-deficient conditions, as indicated by leaf chlorosis, leading to lower chlorophyll content, photosynthetic rate, relative growth rate and then leaf yield harvested. A N-sufficient condition was achieved as the effect of 20 and 40 g N plant^−1^ application, producing a great growth and yield performance due to a high carbon assimilation rate. However, a dose of 40 g N plant^−1^ does not automatically become the best fertilizer recommendation, since 20 g N plant^−1^ is more efficient with a relatively similar effect for increasing kaffir lime leaf production.

## Figures and Tables

**Figure 1 plants-12-01155-f001:**
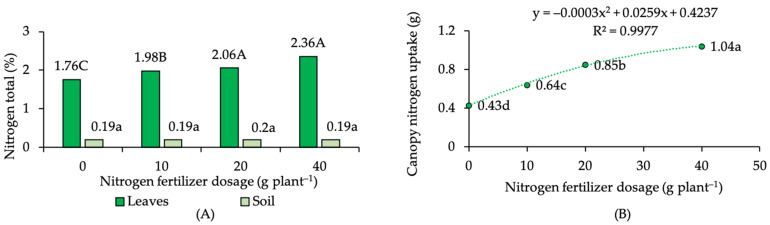
(**A**) Bar chart of nitrogen status on kaffir lime leaves and soil, and (**B**) regression curve of kaffir lime canopy nitrogen uptake under different nitrogen-fertilizer dosage. Note: mean values followed by the same letter are not significantly different based on DMRT at α 0.05.

**Figure 2 plants-12-01155-f002:**
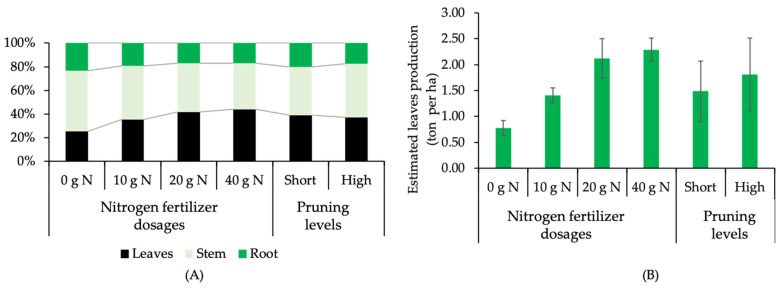
(**A**) Biomass partition and (**B**) estimated leaf production (ton per ha) of kaffir lime under different nitrogen-fertilizer dosages and pruning levels.

**Figure 3 plants-12-01155-f003:**
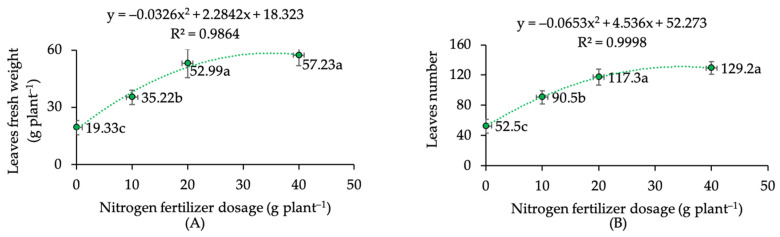
Regression curve of kaffir lime (**A**) fresh weight of leaves and (**B**) leaf numbers under different nitrogen-fertilizer dosages. Note: mean values followed by the same letter within the same curve are not significantly different based on DMRT at α 0.05.

**Figure 4 plants-12-01155-f004:**
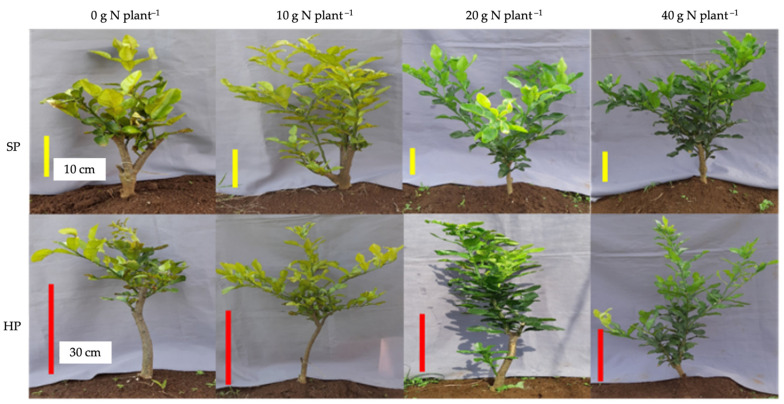
Kaffir lime appearance under different pruning levels and nitrogen-fertilizer dosages.

**Table 1 plants-12-01155-t001:** Pearson correlation analysis between nitrogen status, characteristics of production and physiology of kaffir lime.

	NTS	NLC	CNU	RGR	PH	SN	LN	LFW	PFW	PR	SC	TR	CA	CB	CT
NLC	0.05														
CNU	0.28	0.97													
RGR	0.42	0.93	0.98												
PH	0.57	0.84	0.94	0.97											
SN	0.15	0.98	0.98	0.96	0.87										
LN	0.39	0.93	0.98	0.99	0.94	0.97									
LFW	0.61	0.82	0.92	0.97	0.97	0.87	0.97								
PFW	0.44	0.92	0.98	0.99	0.97	0.95	0.99	0.98							
PR	0.54	0.82	0.92	0.94	0.99	0.84	0.90	0.93	0.94						
SC	0.40	0.27	0.33	0.43	0.30	0.40	0.50	0.53	0.44	0.17					
TR	0.53	−0.79	−0.65	−0.51	−0.40	−0.71	−0.52	−0.30	−0.50	−0.43	0.19				
CA	0.07	0.98	0.97	0.91	0.86	0.96	0.90	0.80	0.91	0.86	0.13	−0.81			
CB	0.05	0.98	0.96	0.91	0.85	0.96	0.89	0.79	0.90	0.85	0.13	−0.82	0.99		
CT	0.06	0.98	0.97	0.91	0.86	0.96	0.90	0.80	0.90	0.86	0.13	−0.81	1.00	0.99	
ANT	−0.77	0.58	0.37	0.24	0.04	0.50	0.28	0.02	0.22	0.04	−0.04	−0.89	0.53	0.55	0.54

Note: NTS—nitrogen total in soil, NLC—nitrogen content in leaves, CNU—canopy nitrogen uptake, RGR—relative growth rate, PH—plant height, SN—shoot numbers, LN—leaf numbers, LFW—fresh weight of leaves, PFW—plant fresh weight, PR—photosynthetic rate, SC—stomatal conductivity, TR—transpiration rate, CA—chlorophyll α, CB—chlorophyll β, CT—chlorophyll total.

**Table 2 plants-12-01155-t002:** Kaffir lime growth performance under different pruning levels and nitrogen-fertilizer dosages.

Treatment	RGR (g week^−1^)	Plant Height (cm)	Shoot Number	Leaf Numbers
Nitrogen fertilizer (N) factor				
0 g N	1.71 ± 0.27 c	53.75 ± 14.75 b	6.50 ± 1.05 b	52.50 ± 9.07 c
10 g N	2.39 ± 0.22 b	58.75 ± 13.65 b	7.50 ± 1.38a b	90.50 ± 8.55 b
20 g N	3.15 ± 0.43 a	79.13 ± 24.91 a	7.83 ± 1.72 a	117.33 ± 10.31 a
40 g N	3.40 ± 0.38 a	79.29 ± 23.27 a	8.67 ± 2.34 a	129.17 ± 8.40 a
Pruning (P) factor				
SP (10 cm)	2.42 ± 0.65 b	51.87 ± 8.95 b	8.92 ± 1.56 a	92.92 ± 32.54 a
HP (30 cm)	2.90 ± 0.78 a	83.59 ± 19.41 a	6.33 ± 0.65 b	101.83 ± 30.69 a
N*P	Ns	Ns	Ns	Ns

Note: mean values within the same column and same factor followed by the same letter are not significantly different based on DMRT at α 0.05. RGR—relative growth rate, SP—short pruning, HP—high pruning, N*P—the interaction of nitrogen-fertilizer dosages and pruning levels, Ns—not significant.

**Table 3 plants-12-01155-t003:** Plant production (fresh weight) of kaffir lime under different pruning levels and nitrogen-fertilizer dosages.

Treatment	Fresh Weight (g)			
	Plant	Leaves	Stem	Root
Nitrogen-fertilizer (N) factor				
0 g N	76.36 ± 13.18 c	19.33 ± 3.65 c	39.29 ± 8. 21 c	17. 75 ± 1.73 c
10 g N	99.10 ± 11.74 b	35.22 ± 3.67 b	44.69 ± 7. 53 b	19. 18 ± 1. 72 bc
20 g N	124.51 ± 20.15 a	52. 99 ± 9. 46 a	51. 94 ± 9. 82 a	21.31 ± 1. 83 ab
40 g N	131.53 ± 16.56 a	57.23 ± 5. 59 a	50. 21 ± 10.91 a	22. 37 ± 1.43 a
Pruning (P) factor				
SP (10 cm)	94.71 ± 20.98 b	37. 12 ± 14. 67 b	38. 46 ± 4. 44 b	19. 13 ± 2. 36 b
HP (30 cm)	121.03 ± 25.94 a	45. 26 ± 17. 66 a	54.61 ± 6.80 a	21. 16 ± 2. 10 a
N*P	Ns	Ns	Ns	Ns

Note: mean values within the same column and same factor followed by the same letter are not significantly different based on DMRT at α 0.05. SP—short pruning, HP—high pruning, N*P—the interaction of nitrogen-fertilizer dosages and pruning levels, Ns—not significant.

**Table 4 plants-12-01155-t004:** Kaffir lime physiological response under different pruning levels and nitrogen-fertilizer dosages.

Treatment	Pn	Tr	Sc	WUE	LUE
Nitrogen-fertilizer (N) factor					
0 g N	17.52 ± 1.19 b	6.54 ± 1.56	0.371 ± 0.09	2.68 ± 0.76	0.58 ± 1.56
10 g N	17.63 ± 1.22 b	6.50 ± 1.85	0.383 ± 0.12	2.74 ± 0.66	0.74 ± 1.85
20 g N	20.03 ± 1.13 a	6.59 ± 1.39	0.381 ± 0.09	3.04 ± 0.81	0.92 ± 1.39
40 g N	20.20 ± 0.99 a	6.25 ± 0.96	0.376 ± 0.06	3.23 ± 1.03	0.98 ± 0.96
Pruning (P) factor					
H10	18.69 ± 1.81	6.70 ± 1.64	0.397 ± 0.10 a	2.81 ± 1. 10	0.71 ± 1.64
H30	18.99 ± 1.59	6.24 ± 1.22	0.358 ± 0.08 b	3.04 ± 1. 30	0.90 ± 1.22
N*P	Ns	Ns	Ns	Ns	Ns

Note: mean values within the same column and same factor followed by the same letter are not significantly different based on DMRT at α 0.05. Pn—photosynthetic rate (μmol CO_2_ m^−2^ s^−1^), Tr—transpiration rate (mmol H_2_O m^−2^ s^−1^), Sc—stomatal conductance (mol H_2_O m^−2^ s^−1^), Sl—stomatal limitation to CO_2_ uptake (mol m^−2^ s^−1^), WUE—intrinsic water-use efficiency (μmol CO_2_ mmol H_2_O^−1^), LUE—instantaneous light-use efficiency (μg lux^−1^). Ns—not significant, N*P—the interaction of nitrogen-fertilizer dosages and pruning levels.

**Table 5 plants-12-01155-t005:** Kaffir lime leaves’ pigment content under different nitrogen-fertilizer dosages.

Treatment	Chlorophyll α (mg g^−1^)	Chlorophyll β (mg g^−1^)	Chlorophyll Total (mg g^−1^)	Anthocyanin (mg 100 g^−1^)
0 g N	0.319 ± 0.12 d	0.122 ± 0.06 d	0.441 ± 0.17 d	0.063 ± 0.04 a
10 g N	0.618 ± 0.08 c	0.229 ± 0.02 c	0.846 ± 0.10 c	0.077 ± 0.01 a
20 g N	0.953 ± 0.06 b	0.337 ± 0.02 b	1292 ± 0.09 b	0.051 ± 0.01 a
40 g N	1.687 ± 0.12 a	0.603 ± 0.04 a	2.291 ± 0.16 a	0.088 ± 0.01 a

Note: mean values within the same column followed by the same letter are not significantly different based on DMRT at α 0.05.

## Data Availability

Not applicable.

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
