# Peer review of "Agro-Physiological Traits of Kaffir Lime in Response to Pruning and Nitrogen Fertilizer under Mild Shading"

_plants, 2023, doi:10.3390/plants12051155_

Round 1

Reviewer 1 Report (Previous Reviewer 1)

Comments:

1.       Abstract: I would like to suggest adding more robust take-home messages about Kaffir Lime in the abstract.

2.       Introduction: Authors improved missing points in introduction however I would recommend them to improve justification for those missing knowledge gaps throughout the introduction.

3.       Tables: Please either remove letters or standard deviations from the tables. There is no need to keep both since they give same information.

4.       Discussion: I would recommend improving the discussion with involving more data from results. Discussion seems weaker compared to results section.

Author Response

Reviewer 2 Report (Previous Reviewer 2)

The paper after revision is significantly improved and can be send to production department

Author Response

Reviewer 3 Report (Previous Reviewer 3)

The authors did not address my major concern which I mentioned in my last review report. 

Round 2

Reviewer 3 Report (Previous Reviewer 3)

The authors clearly explained and addressed my comments. I recommend the revised manuscript for publication.  

This manuscript is a resubmission of an earlier submission. The following is a list of the peer review reports and author responses from that submission.

Round 1

Reviewer 1 Report

Please find attached file for comments.

Reviewer 2 Report

Review Plants - 2155854

Determination of Harvest-Pruning Type and Nitrogen Dose and based on Agro-Physiological Traits in Leaf Oriented-Minor Citrus of Kaffir Lime under Artificial Shading

The title is too long and incomprehensible. Similarly incomprehensible is the abstract. The sentences are too long and their construction makes them impossible to understand.

Please decide it is harvest or pruning, but not harvest-pruning. These activities have different purposes. If the goal is to harvest leaves, then of course you need to harvest by pruning or cutting the plants.

Lines 33-38: Statements that are debatable to say the least. Please read the work: Curk et al.(2016).  Phylogenetic origin of limes and lemons revealed by cytoplasmic and nuclear markers. Annals of Botany.11 (4): 565–583. doi:10.1093/aob/mcw005

Lines 75-82: No clearly defined research purpose or objectives.

...... unintelligible

Lines 322-323: it is not clear. Plants were produced as a seedling or were grafted? How rootstock was propagated. If as a seedling, how was the differentiation of growth force avoided?

Line 329: Nested design? unintelligible

Lines 335-336: what is replication and what is experimental unit? How many plants were planted?

Line 339: The field was transplanting?

Line 343: Rootstock variety? What the variety is rootstock?

Line 344: What is hoarding technique. What do you collect?

Most of paragraph 4.3. and 4.4 needs to be rewritten. All measurements should be discussed separately and the procedure described more precisely.

Line 386: Present experiment success to determine the best practice? unintelligible

Reviewer 3 Report

Manuscript title: Determination of Harvest-Pruning Type and Nitrogen Dose 2 and based on Agro-Physiological Traits in Leaf Oriented-Minor 3 Citrus of Kaffir Lime under Artificial Shading

Comments to authors:

Budiarto et al, investigated the impact of harvest pruning type and nitrogen fertilizer on citrus lime growth characteristics. The topic is interesting, and the authors presented it in a good manner. It is surprising to me that the interactive effect of pruning type and nitrogen doses is non-significant for all studied parameters. If this is the case, one can assume that high-pruning type plants had more assimilates therefore plant growth was better regardless of nitrogen doses. Clearly explain this interactive effect in your discussion. My minor suggestions are given below.

From the title, remove ‘and’ before …based on

Line 20: add the scientific name of Rangpur lime

Line 24: what is (i)?

In abstract: is there any difference between r and R2?

In lines 25-27: what is the recommendation about pruning?

Line 202-21: what are the levels of pruning and N doses?

Line 28: remove N-deficient and N-sufficient.

Line 317: replace ‘and’ with ‘to’

Line 226: there should be ‘is’ instead of
